# Seroprevalence Rates against West Nile, Usutu, and Tick-Borne Encephalitis Viruses in Blood-Donors from North-Western Romania

**DOI:** 10.3390/ijerph19138182

**Published:** 2022-07-04

**Authors:** Mircea Coroian, Andrei Daniel Mihalca, Gerhard Dobler, Kathrin Euringer, Philipp Girl, Silvia-Diana Borșan, Zsuzsa Kalmár, Violeta Tincuța Briciu, Mirela Flonta, Adriana Topan, Amanda Lelia Rădulescu, Andrei Ungur, Mihaela Sorina Lupșe

**Affiliations:** 1Department of Parasitology and Parasitic Diseases, University of Agricultural Sciences and Veterinary Medicine of Cluj-Napoca, 400372 Cluj-Napoca, Romania; mircea.coroian@usamvcluj.ro (M.C.); amihalca@usamvcluj.ro (A.D.M.); silvia.borsan@usamvcluj.ro (S.-D.B.); 2Department of Infectious Diseases, University of Medicine and Pharmacy “Iuliu Hațieganu” Cluj-Napoca, 400012 Cluj-Napoca, Romania; briciu.tincuta@umfcluj.ro (V.T.B.); topanadriana@yahoo.com (A.T.); aradulescu@umfcluj.ro (A.L.R.); mihaela.lupse@yahoo.com (M.S.L.); 3Bundeswehr Institute of Microbiology, 85748 Munich, Germany; gerharddobler@bundeswehr.org (G.D.); kathrineuringer@bundeswehr.org (K.E.); philipp.girl@bundeswehr.org (P.G.); 4German Centre for Infection Research (DZIF), 85748 Munich, Germany; 5Department of Microbiology, Immunology and Epidemiology, University of Agricultural Sciences and Veterinary Medicine of Cluj-Napoca, 400372 Cluj-Napoca, Romania; 6Hospital for Infectious Diseases, 400348 Cluj-Napoca, Romania; m.flonta@yahoo.com; 7Department of Pathology, University of Agricultural Sciences and Veterinary Medicine of Cluj-Napoca, 400372 Cluj-Napoca, Romania; andrei.ungur@usamvcluj.ro

**Keywords:** arboviruses, public health, tick-borne encephalitis virus, Usutu virus, vector-borne diseases, West Nile virus

## Abstract

Introduction: West Nile virus (WNV), Usutu virus (USUV), and the tick-borne encephalitis virus (TBEV) are all arboviruses belonging to *Flaviviridae* family. All are characterized by vectorial transmission and sometimes associated with neuroinvasive infections. The circulation of these viruses is considered endemic in parts of Europe, with human cases reported in many countries. Among hosts, the viruses are vectored by hematophagous arthropods, such as mosquitoes (WNV, USUV) and ticks (TBEV). Considering the currently outdated knowledge regarding the epidemiology of these viruses in Romania, the aim of our study was to assess the seroprevalence rates of WNV, USUV, and TBEV among healthy blood donors in north-western Romania. Methods: Human blood samples from healthy donors were collected between November 2019 and February 2020 in six counties from the north-western region of Romania. The samples were serologically tested by ELISA and serum neutralization test. Results: Overall, we obtained a seroprevalence of 3.17% for WNV, 0.08% for TBEV, and 0% for USUV. Conclusion: Despite the low seroprevalence of WNV, USUV, and TBEV in our study, we highlight the need for continuous nationwide vector and disease surveillance and implementation of control measures. Further research is required for an optimal overview of the epidemiological status of the Romanian population regarding these flaviviruses together with countrywide awareness campaigns.

## 1. Introduction

West Nile virus (WNV), Usutu virus (USUV), and the tick-borne encephalitis virus (TBEV) are all arboviruses belonging to *Flaviviridae* family, characterized by vectorial transmission and sometimes associated with neuroinvasive infections. The circulation of these viruses is considered endemic in some parts of Europe, with cases reported in many countries. Hematophagous arthropods such as mosquitoes (WNV, USUV) and ticks (TBEV) transmit the virus among hosts [1].

Romania has a long history regarding the circulation of the WNV, the disease being first reported in 1955 [2]. In addition, during 1996, Romania also recorded the most important human outbreak reported in Europe at the time, in the south-eastern region of the country. Of the 393 human cases recorded, 352 were severe forms of meningoencephalitis [3]. A continued transmission was observed during the next years but with a lower number of clinical infections [4]. Another notable outbreak occurred during 2010, with 52 confirmed cases and a 10% mortality rate. Although most infections were located in the southern part of the country, new cases have also been reported inside the arch of the Carpathian Mountains [5]. Moreover, during 2016, another disease outbreak was recorded in Romania, this time registering 93 neurological human cases [6,7]. To date, the most severe outbreak registered in Europe in recent years took place in 2018, when Romania registered 277 clinical cases and 43 deaths out of a total of 2083 human clinical infections at the European level [8]. A decrease in infections was observed in the following years [9,10,11]. Since 1997, a passive surveillance system has been implemented in Romania. Every year, from June to November, blood serum and cerebrospinal fluid from suspect cases of human WNV-associated central nervous system infections in patients over the age of 15 years old are screened using IgM WNV enzyme-linked immunosorbent assay (ELISA). A 28-day quarantine period is mandatory for all blood donors from localities where human cases have been detected [12]. Nevertheless, a significant number of human viral encephalitis cases remain unconfirmed for WNV and are recorded as “viral encephalitis with unknown etiology”, as the current legislation does not require further confirmatory tests for other insect-borne arboviruses. As a consequence, the blood donors are not screened for other viruses although some, i.e., Toscana [13] and Usutu [14], were recently reported in Romania.

USUV, a member of the Japanese encephalitis serocomplex, is phylogenetically close to WNV [15]. USUV has spread to a large part of the European continent over the two decades, mainly leading to substantial avian mortalities with a significant recrudescence of bird infections recorded throughout Europe within the last few years [16]. In Europe, USUV was first reported in Austria and was associated with high mortality among blackbirds (*Turdus merula*) [17]. This event was followed by a retrospective study in Italy on bird tissue samples stored from 1996, which subsequently tested positive for USUV by molecular techniques [18]. Later, the virus was identified in mosquitoes and different vertebrate hosts in several European countries [19,20]. USUV infection in humans is considered to be most often asymptomatic or to cause mild clinical signs [16]. The first human neuroinvasive infection in Europe was registered in 2009 in Italy in a patient with meningoencephalitis symptoms [21] followed by other reports [22,23,24]. Seroconversion in healthy blood donors was also registered [25,26,27,28]. In Romania, the first serological evidence of the presence of USUV was documented in a domestic dog [14], but its presence in humans has not yet been demonstrated.

TBEV is endemic in many European countries including Romania, being the most important neuroinvasive arbovirus vectored by ticks [29,30,31]. Information on the tick-borne encephalitis (TBE) epidemiology in Romania is scarce and partly outdated. The most important outbreak in humans was recorded in 1999, when 38 infections were recorded, raw goat dairy products being incriminated as the source of infection [32]. Seroprevalence rates varied between 0.0% and 41.5% in humans and between 0.0% and 27.7% in livestock [33,34,35]. Since 2008, TBE has been passively monitored in 11 north-western and central counties considered at risk out of the 41 counties of Romania [36]. TBE is also notifiable at the EU level since 2012 [37].

Considering the outdated current knowledge on the epidemiology of these viruses in Romania, this study aimed to assess the seroprevalence of WNV, USUV, and TBEV among healthy blood donors in north-western Romania.

## 2. Materials and Methods

### 2.1. Sample Collection

Human blood samples from healthy donors were collected between November 2019 and February 2020 and between August and September 2020 by the Regional Blood Transfusion Centers in six counties from the north-western region of Romania (Alba (AB), Cluj (CJ), Sălaj (SJ), Bistrița-Năsăud (BN), Maramureș (MM), and Satu-Mare (SM)), as previously described [38]. Two hundred samples were analyzed per county, resulting in a total of 1200 samples being assessed in the current study. Each donor answered a questionnaire survey concerning age, gender, occupation, and living environment. The respondents were divided into 3 groups according to their age as follows: young adults (18 ≤ ages ≤ 35), middle-aged (36 ≤ ages ≤ 55), and old adults (ages ≥ 56). Additionally, in relation to job profile, the participants were grouped by higher and secondary education level, respectively, with predominantly indoor/outdoor activity.

After collection, each sample was centrifuged at 8000 rpm for 10 min and stored at −80 °C until further analysis.

### 2.2. Ethical Statement

The study was approved by the National Institute of Hematology and Blood Transfusion of Romania (Registration Number: 2589/c/24 October 2019). All patients offered their informed consent prior to sample collection.

### 2.3. Serological Analysis

#### 2.3.1. ELISA

All serum samples were screened in the Laboratory of the Clinical Hospital for Infectious Diseases from Cluj-Napoca using a commercial enzyme-linked immunosorbent assay (ELISA) for WNV (Euroimmun IgG West Nile—Medizinische Labordiagnostika AG, Lübeck, Germany) and TBEV (SERION ELISA classic FSME/TBE Virus IgG—Serion Diagnostics, Wurzburg, Germany) IgG antibodies. Both tests were performed according to the manufacturer’s instructions.

#### 2.3.2. Serum-Neutralization Tests

All neutralization tests were performed as micro serum neutralization (micro-SNT) tests according to standard procedures [39] using the validated protocol of the accredited diagnostic laboratory at the Bundeswehr Institute of Microbiology, Munich. The described tests for the particular flaviviruses were conducted in parallel, modified only in using the respective mentioned flavivirus strains and the respective cell lines. Therefore, the titers against all three tested flaviviruses were comparable with each other.

##### WNV

WNV (strain EgAn101; kindly provided by Robert Shope Yale Arbovirus Research Unit) was cultured in Vero cells, and virus stocks (titrated to 40–60 TCID/50 µL) were prepared and stored at −80 °C until further use. The SNTs were screened in the Bundeswehr Institute of Microbiology, Munich, Germany, and performed in 96-well cell culture plates (Greiner bio-one, Frickenhausen, Germany). Patient sera were inactivated at 56 °C for 30 min and then diluted two-fold in duplicate beginning with 1:20 to 1:2560 in Minimal Essential Medium (MEM, plus MEM Non-Essential Amino Acids Solution plus Antibiotic-Antimycotic Solution; all Invitrogen, ThermoFisher Scientific, Darmstadt, Germany). A cell control and a virus re-titration were used as controls on each 96-well plate. The respective virus (40–60 TCID50) was added to each well, and the serum–virus solution was incubated for one hour at 37 °C (5% CO_2_). Afterward, Vero B4 cells (1 × 10^4^ cells/50 µL) were added to each well and incubated for 5–7 days at 37 °C. The supernatants were then discarded, and the 96-well plates were fixed in 13% formalin/PBS and stained with crystal violet (0.1%) and titers visually determined.

The antibody titer corresponding to the highest serum dilution showing complete inhibition of cytopathic effect (CPE) in both wells were reported. Due to a shortage of serum available, the 1:20 starter dilution was chosen. If only one of the wells of the 1:20 solution showed neutralizing capacity, the titer for the serum concerned was stated as 1:10. Thus, the samples were classified as either “NT-negative” (titer < 1:10) or “NT-positive” (titer ≥ 1:10), with the highest readable titer being ≥1:2560.

##### TBEV

The TBEV-SNT was performed in the Bundeswehr Institute of Microbiology, Munich, Germany, as previously described [40]. The virus strain used was TBEV strain Neudörfl with 100 TCID50 per test, and A549 cells were used for the test.

##### USUV

To exclude the cross-reactions, the samples were also tested for USUV by SNT. The USUV-SNT was performed in the Bundeswehr Institute of Microbiology, Munich, Germany, according to the micro-SNT described. The virus strain used was kindly provided by Martin Pfeffer with 100 TCID50 per test, and Vero B4 cells were used. The fourfold or higher difference in NT antibody titers of a particular serum between flaviviruses and, here, USUV and WNV is generally accepted as specific for the respective flavivirus [41].

### 2.4. Statistical Analysis

The statistical analysis was performed using Epi InfoTM 2000 software (https://www.cdc.gov/epiinfo, accessed on 17 February 2021). We used the infection prevalence and the 95% confidence interval in statistical calculations (*p*-values < 0.05 were considered significant). Continuous normally distributed variables are reported as median and interquartile range, and categorical variables are presented as frequencies and percentages.

## 3. Results

### 3.1. Study Group

The study group consisted of 1200 samples collected from healthy blood donors in six counties from the north-western region of Romania (200 samples from each county), as previously described [38]. Briefly, the age of the study participants varied between 18 and 65 years, with a median age of 41 years (interquartile range 53–29). Group characteristics concerning the gender, age category, education level, activities, and living environment are described in Table 1.

### 3.2. ELISA

Of the 1200 samples screened for WNV by ELISA, 3.3% (39/1200; 95% CI 2.4–4.4) showed positive results, while 0.3% (3/1200; 95% CI 0.1–0.7) had equivocal antibody index values, whereas 2.3% (28/1200; 95% CI 1.6–3.4) of the samples had positive IgG ELISA index, and 1.3% (16/1200; 95% CI 0.8–2.2) had equivocal IgG ELISA index for TBEV. Twenty-six samples (2.2%) had positive results for both WNV and TBEV (Table 2).

### 3.3. SNT

Serum samples from the donors with positive or equivocal IgG ELISA index values for WNV (3.5%; 42/1200; 95% CI 2.6–4.7) and for TBEV (3.7%; 44/1200; 95% CI 0.0–0.5) were further analyzed by SNT.

Overall, 3.2% (38/1200; 95% CI 2.3–4.3) of the samples had positive SNT results for WNV and 0.1% (1/1200; 95% CI 0.1–0.5) for TBEV. Thirty-four (2.8%; 95% CI 2.0–3.9) ELISA-positive and two ELISA-equivocal (0.2%; 95% CI 0.1–0.5) serum samples were confirmed.

Among the 28 (2.3%; 95% CI 1.6–3.4) ELISA-positive samples for TBEV, 1 (0.1%; 95% CI 0.1–0.5) were positive by SNT, and none of the serum samples with equivocal ELISA result (1.3%, 16/1200; 95% CI 0.8–2.2), had positive SNT results (Figure 1).

All the samples tested for USUV-SNT returned negative results. Therefore, it was proven that USUV was not responsible for the positive results, and there was no cross-reactivity in the tested sera due to this virus.

### 3.4. WNV

Higher seroprevalence was found in males (4.0%) than in females (1.5%) and in the donors with secondary education (4.3%) relative to those with higher education (1.9%). These differences were statistically significant (Table 3). While 4.1% of the donors from rural environments had positive SNT results, the seroprevalence of blood donors from urban environments was 2.7%. According to their occupational fields, 4.9% of the donors with outdoor activities and 2.9% with indoor activities had positive SNT results (Table 3).

For each age category, the seroprevalence was higher in men than in women, and the highest SNT rates were obtained in the old adults (6.0%) followed by middle-aged (3.3%) and young adults (2.8%), with statistically significant results for the middle-aged category (Table 4).

Nevertheless, statistically significant results were registered between the counties (χ^2^ = 51.6891, df = 10; *p* = 0.001), and the seroprevalence varied between 0.5% (Cluj-Napoca) and 10.5% (Satu-Mare) in the investigated counties (Table 5). We registered statistically significant results within the counties for young adults (χ^2^ = 21.5796, df = 5; *p* = 0.0006) and, respectively, for the middle-aged (χ^2^ = 19.4194, df = 5; *p* = 0.0016) group.

### 3.5. TBEV

The only TBEV positive sample by SNT was collected from the middle-aged category: a 44-year-old male from Satu-Mare county (Table 5) who also tested positive for WNV by SNT. He was living in a rural environment with predominant indoor activity and secondary education level (Table 3).

## 4. Discussion

The WNV, TBEV, and USUV are arthropod-transmitted flaviviruses widespread throughout Europe. A common trait of these viruses is their cross-reactivity encountered in serological tests, which can sometimes pose diagnostic challenges. Moreover, the national regulations for the surveillance of these viruses are highly heterogenic throughout Europe [42,43].

WNV is the most prevalent mosquito-borne virus reported in Romania, with human infections recorded yearly [44]. It is noteworthy that the herein study was performed in an area of Romania (north-west) considered at low risk for WNV [45]. Nevertheless, the seroprevalence obtained in this study is similar to the values recorded in Bucharest during the 1996 outbreak [46]. This could suggest the continuous circulation of the virus in the population and a high number of asymptomatic cases. Moreover, a recent study conducted in Romania including 176 samples collected from inhabitants in Iași county (north-east) revealed a WNV seroprevalence of 3.4% [14]. The seroprevalence rates in our study are higher than those reported in blood donors in Italy (0.68%, 0.61%, 0.78%) and Hungary (2.19%) [47,48,49,50]. The high seroprevalence rate registered in Satu-Mare and the low seroprevalence from the other counties closely follow the risk model previously published by our team, who identified the human population, precipitation, and altitude as the most important factors in predicting the West Nile virus infections in humans [45].

As we only considered positive the results confirmed by SNT, the discussion section is focused on these results. Contrary to prior studies, the gender of the participants was identified as a risk factor in the current research [14,47,51], and higher seropositivity among males can be associated with more time spent outdoors and higher exposure to mosquito bites. Moreover, our analysis did not identify this occupational hazard as a risk factor in acquiring the virus, as previously reported in Romania [52]. Previous studies suggested that the higher positivity rate in the rural environments could be influenced by weaker measures of surveillance and control in these sites and also by the amount of time spent outdoors performing agricultural activities [53]. However, we found no statistically significant differences between urban and rural cohorts. Other studies suggested higher seroprevalence rates of WNV in the elderly related to increased outdoor time and less frequent use of repellents [51,53]. Nevertheless, we did not observe this correlation in the present study. Additionally, we observed that the lower education level of the studied population was linked to a higher WNV exposure, as suggested by Hadjichristodoulou et al. (2015) [51], possibly due to inadequate knowledge of mosquito control and less use of insect repellents.

USUV is generally less commonly diagnosed compared to WNV, and several studies highlighted their co-circulation in vectors and hosts [54,55]. Hence, the negative results for USUV could potentially be correlated with the low prevalence of WNV and might suggest the absence of virus circulation in the north-west region of Romania. Interestingly, a recent study reported antibodies against USUV in a dog from Iași [14], a risk area for WNV infection in Romania [45]. Serological surveys conducted on blood donors in several countries in Europe revealed prevalence rates between 0% to 6.57% in humans [26,48,49,56].

The present study depicts a rather low seroprevalence rate regarding the TBEV in the assessed samples compared to countries where it is considered endemic [57,58,59], yet it is comparable with results obtained by Christova et al. (2017) in Bulgaria (0.6%) [60]. Low seroprevalence rates were also reported in studies conducted on healthy blood donors in Norway (0.65, 0.4%) [61,62].

Because no neutralization methods were used in prior studies conducted in Romania [33,34], the indicated higher seroprevalence rates should be cautiously interpreted.

Nonetheless, it is important to mention that our sampling protocol had its limitations. The sampling protocol did not include questions on history of mosquito/tick bites, previous symptoms of central nervous system infections (CNS)/documented CNS infections of viral etiology, occupational risk, or vaccination history for flaviviruses. The diagnosis and immunity testing of flaviviruses should always include an evaluation of immune responses against different flaviviruses such as TBEV, WNV, yellow fever virus, Japanese encephalitis virus, and dengue viruses. Regarding TBEV vaccination, it is important to outline that FSME-IMMUN^®^ (Pfizer), the only vaccine available in Romania, received authorization for use in our country in 2019, and no official recommendations for use in general population/specific groups or funding by the National Health system for use are available. As the primary vaccination includes three doses in a minimum 6-month interval, and the study group investigated in November 2019–September 2020, we do not expect to have an important number of TBE-vaccinated patients in our group. Cross-reactivity might be associated with the yellow fever vaccine, but neutralizing antibodies are thought to be the most specific antibodies produced by the host and with the lowest cross-reactivity to other flaviviruses [63]. On the other hand, yellow fever vaccination is an extremely rarely administered vaccine in Romania and performed only in travel-medicine clinics. Nevertheless, only the history of the patient together with the serological results against the most common flaviviruses and flavivirus vaccinations will give a realistic picture of the immune status and of a potential infection.

To our best knowledge, this is the first comprehensive study that aimed to analyze the seroprevalence of WNV among blood donors in Romania and the first large-scale study that evaluates the presence of the Usutu virus in the human population in Romania. Moreover, there are no recent studies on the presence of TBEV in Romania. Thus, the research herein aimed to update the existing data and bring additional information regarding this aspect.

## 5. Conclusions

Despite the low seroprevalence of WNV, USUV, and TBEV in our study, we highlight the need for continuous nationwide vector and disease surveillance and implementation of control measures. Further research is required for an optimal overview of the epidemiological status of the Romanian population regarding these flaviviruses together with countrywide awareness campaigns.

## Figures and Tables

**Figure 1 ijerph-19-08182-f001:**
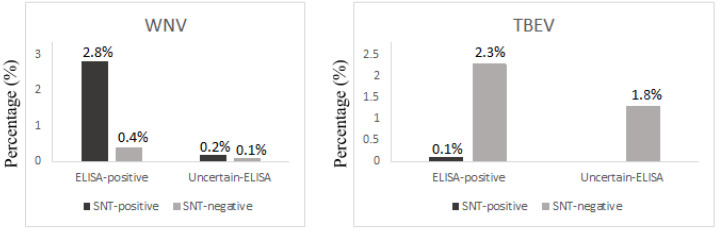
SNT results for confirmation of ELISA-positive or ELISA-equivocal IgG results.

**Table 1 ijerph-19-08182-t001:** Group characteristics concerning variables.

Variable		% (*n*; 95% CI)
Gender	Males	66.2 (794; 63.4–68.8)
	Females	33.8 (406; 31.2–36.6)
Age category	Young adults	47.4 (569; 44.6–50.3)
	Middle-aged	48.4 (581; 45.6–51.2)
	Old adults	4.2 (50; 3.2–5.5)
Education level	Higher	45.1 (541; 42.3–47.9)
	Secondary	54.9 (659; 52.1–57.7)
Activities	Outdoor	11.9 (143; 10.2–13.9)
	Indoor	88.1 (1057; 86.1–89.8)
Environment	Urban	67.4 (809; 64.7–70.0)
	Rural	32.6 (391; 30.0–35.3)

*n*, total number of samples; %, prevalence; 95% CI, confidence interval.

**Table 2 ijerph-19-08182-t002:** IgG ELISA results for WNV and TBEV.

ELISA	TBE		
WNV	Positive	Equivocal	Negative
	% (*n*; 95% CI)		
Positive	2.2 (26; 1.5–3.2)	0.4 (5; 0.2–1.0)	0.7 (8; 0.3–1.3)
Equivocal	0.0	0.1 (1; 0.1–0.5)	0.2 (2; 0.1–0.6)
Negative	0.2 (2; 0.1–0.6)	0.8 (10; 0.5–1.5)	95.5 (1146; 94.2–96.5)

*n*, total number of samples; %, seroprevalence; 95% CI, confidence interval.

**Table 3 ijerph-19-08182-t003:** WNV and TBEV seroprevalence by SNT according to gender, environment, education level, activities, and age group category.

Variables	WNV		TBEV	
	% (+/*n*; 95% CI)	*p*	% (+/*n*; 95% CI)	*p*
Gender				
Females	1.5 (6/406; 0.7–3.2)	**0.0267**	0 (0/406)	0.4743
Males	4.0 (32/794; 2.9–5.6)		0.1 (1/794; 0.1–0.7)	
Environment				
Urban	2.7 (22/809; 1.8–4.1)	0.2727	0 (0/809)	0.7100
Rural	4.1 (16/391; 2.5–6.5)		0.3 (1/391; 0.1–1.4)	
Education				
Higher	1.9 (10/541; 1.1–3.4)	**0.0280**	0 (0/541)	0.3647
Secondary	4.3 (28/659; 3.0–6.1)		0.2 (1/659; 0.1–0.9)	
Activities				
Outdoor	4.9 (7/143; 2.0–9.8)	0.3157	0 (0/143)	0.7128
Indoor	2.9 (31/1057; 212–4.1)		0.1 (1/1057; 1.5–3.4)	
Age group				
Young adults	2.8 (16/569; 1.7–4.5)	0.4578	0 (0/569)	0.5868
Middle-aged	3.3 (19/581; 2.1–5.1)		0.2 (1/581; 1.3–3.8)	
Old adults	6.0 (3/50; 1.3–16.6)		0 (0/50)	
**Total**	**3.2 (38/1200; 2.3–4.3)**		**0.1 (1/1200; 0.0–0.5)**	

%, seroprevalence; +/*n*, number of positive or equivocal samples/total number of samples; CI, confidence interval, statistically significant *p* values are bolded.

**Table 4 ijerph-19-08182-t004:** WNV and TBEV seroprevalence by SNT according to age group and gender.

Age Group	Gender	WNV		TBEV	
		% (+/*n*; 95% CI)	*p*	% (+/*n*; 95% CI)	*p*
Young adults	Males	2.9 (11/384; 1.6–5.1)	1.000	0.0 (0/384)	1.000
	Females	2.7 (5/185; 0.9–6. 2)		0.0 (0/185)	
	Total	2.8 (16/569; 1.7–4.5)		0.0 (0/569)	
Middle-aged	Males	4.8 (18/373; 3.1–7.5)	**0.0098**	0.3 (1/373; 0.1–1.5)	1.000
	Females	0.5 (1/208; 0.1–2.7)		0.0 (0/208)	
	Total	3.3 (19/581; 2.1–5.1)		0.2 (1/581; 0.1–1.0)	
Old adults	Males	8.1 (3/37; 1.7–21.9)	0.7038	0.0 (0/37)	1.000
	Females	0 (0/13)		0.0 (0/13)	
	Total	6.0 (3/50; 1.3–16.6)		0.0 (0/50)	
**Total**		**3.2 (38/1200; 2.3–4.3)**	**0.4578**	**0.1 (1/1200; 0.1–0.5)**	**0.5868**

%, seroprevalence; +/*n*, number of positive or equivocal samples/total number of samples; CI, confidence interval: statistically significant *p* values are bolded.

**Table 5 ijerph-19-08182-t005:** WNV and TBEV IgG seroprevalence by SNT in each county for each age category.

County	Age Group	WNV		TBEV	
		% (+/*n*; 95% CI)	*p*	% (+/*n*; 95% CI)	*p*
Alba	Young	4.5 (4/89; 1.2–11.1)	0.7294	0.0 (0/89)	1.0000
	Middle	2.9 (3/104; 0.6–8.2)		0.0 (0/104)	
	Old	0.0 (0/7)		0.0 (0/7)	
	Total	3.5 (7/200; 1.4–7.1)		0.0 (0/200)	
Bistrița-Năsăud	Young	1.2 (1/87; 0.1–6.2)	0.9627	0.0 (0/87)	1.0000
	Middle	0.9 (1/108; 0.1–5.1)		0.0 (1/108)	
	Old	0.0 (0/5)		0.0 (0/5)	
	Total	1.0 (2/200; 0.1–3.6)		0.0 (0/200)	
Cluj	Young	0.7 (1/139; 0.1–3.9)	0.8021	0.0 (0/139)	1.0000
	Middle	0.0 (0/57)		0.0 (0/57)	
	Old	0.0 (0/4)		0.0 (0/4)	
	Total	0.5 (1/200; 0.1–2.8)		0.0 (0/200)	
Maramureș	Young	2.0 (2/100; 0.2–7.0)	0.8022	0.0 (0/100)	1.000
	Middle	3.2 (3/94; 0.7–9.0)		0.0 (0/94)	
	Old	0.0 (0/6)		0.0 (0/6)	
	Total	2.5 (5/200; 0.8–5.7)		0.0 (0/200)	
Satu-Mare	Young	11.3 (7/62; 4.7–21. 9)	0.7743	0.0 (0/62)	0.7001
	Middle	9.4 (11/117; 4.8–16.2)		0.9 (1/117; 0.1–4.7)	
	Old	14.3 (3/21; 3.1–36.3)		0.0 (0/21)	
	Total	10.5 (21/200; 6.6–15.6)		0.5 (1/200; 0.1–2.8)	
Sălaj	Young	1.1 (1/92; 0.1–5.9)	0.9618	0.0 (0/92)	1.0000
	Middle	1.0 (1/101; 0.1–5.4)		0.0 (0/101)	
	Old	0.0 (0/7)		0.0 (0/7)	
	Total	1. 0 (2/200; 0.1–3.6)		0.0 (0/200)	
**Total**		**3.2 (38/1200; 2.3–4.3)**		**0.1 (1/1200; 0.1–0.5)**	

%, seroprevalence; +/*n*, number of positive or equivocal samples/total number of samples; CI, confidence interval.

## Data Availability

All data generated or analyzed during this study are included in this published article. Other datasets used and/or analyzed can be made available by the corresponding author on reasonable request.

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
