# Peer review of "Seroprevalence Rates against West Nile, Usutu, and Tick-Borne Encephalitis Viruses in Blood-Donors from North-Western Romania"

_ijerph, 2022, doi:10.3390/ijerph19138182_

Round 1
Reviewer 1 Report
Review of the manuscript “Seroprevalence rates against West Nile, Usutu and Tick-Borne Encephalitis Viruses in Blood-Donors from North-Western Romania.”.
This manuscript studies the seroprevalence of WNV; USUV and TBEV in the human population from North-Western Romania.
In general, the article is interesting and provides knowledge of the seroprevalence of these viruses in an area of Romania where different outbreaks have occurred. In addition, the authors did a statistical analysis in order to identify differences between age groups or occupational fields.
Major comments.
1) Although the introduction is well written and easy to follow, the main concern is the material and methods section. I suggest to the authors include more information about the Serum-Neutralization tests. Unless to indicate references to the protocol.
2) The authors indicate, that they start in dilution at 1:20 and if only one of the wells at 1:20 showed neutralizing capacity, the titer was stated as 1:10. I think that the authors could lose information and potential positive samples that could show neutralizing capacity at 1:10 but not at 1:20.
3) Another important issue is how the authors did the SNT for TBEV and USUV, it was the USUV SNTs running in parallel to WNV to exclude cross-reactions? How are the results between WNV and USUV compared? What criterium was used? I suggest to the authors include more information about this issue.
4) The SNT results are difficult to follow. I suggest to the author include a different paragraphs for each virus and include more information about USUV and how it was discarded from cross-reaction with other viruses.
5) Improve Figure 1.
6) Tables 3, 4, and 5 change to bold type the significant p-value.
Author Response
Reviewer 1 (R1)
R1: This manuscript studies the seroprevalence of WNV; USUV and TBEV in the human population from North-Western Romania.
In general, the article is interesting and provides knowledge of the seroprevalence of these viruses in an area of Romania where different outbreaks have occurred. In addition, the authors did a statistical analysis in order to identify differences between age groups or occupational fields.
Major comments.
- Although the introduction is well written and easy to follow, the main concern is the material and methods section. I suggest to the authors include more information about the Serum-Neutralization tests. Unless to indicate references to the protocol.
Author response (AR): A reference with the description of the method was added (Van Maanen C, Terpstra C. Comparison of a liquid-phase blocking sandwich ELISA and a serum neutralization test to evaluate immunity in potency tests of foot-and-mouth disease vaccines. Journal of Immunological Methods 124 (1989), 111-119). The following paragraph was added for clarification: “All neutralization tests were performed as micro serum neutralization (micro-SNT) tests according to standard procedures [39] using the validated protocol of the accredited diagnostic laboratory at the Bundeswehr Institute of Microbiology, Munich. The described tests for the particular flaviviruses were conducted in parallel, modified only in using the respective mentioned flavivirus strains and the respective cell lines. Therefore, the titers against all three tested flaviviruses were comparable with each other”.
R1: 2) The authors indicate, that they start in dilution at 1:20 and if only one of the wells at 1:20 showed neutralizing capacity, the titer was stated as 1:10. I think that the authors could lose information and potential positive samples that could show neutralizing capacity at 1:10 but not at 1:20.
AR: The aim of the neutralization test performed was not to gain higher sensitivity. The aim of the NT tests was to differentiate between the particular flavivirus antibodies. Due to the procedure as used in the study starting with 1:20 assured that the fourfold titer difference could be shown (i.e. 1:20 vs. <1:10).
R1: 3) Another important issue is how the authors did the SNT for TBEV and USUV, it was the USUV SNTs running in parallel to WNV to exclude cross-reactions? How are the results between WNV and USUV compared? What criterium was used? I suggest to the authors include more information about this issue.
AR: The point was clarified. An additional reference clarifying the cross-reactivity problem between flaviviruses in polyclonal sera was added (Calisher CH, Karabatsos N, Dalrymple JM, Shope RE, Porterfield JS, Westaway EG, Brandt WE. Antigenic relationship between flaviviruses as detemrined by cross-neutralization tests with polyclonal antisera. Journal of General Virology 70 (1989), 37-43.).
The following text was added for clarification in chapter 2.5.3: “To exclude the cross-reactions, the samples were also tested for USUV by SNT. The USUV-SNT was performed in the Bundeswehr Institute of Microbiology, Munich, Germany, according to the micro-SNT described. The virus strain used was kindly provided by Martin Pfeffer with 100 TCID50 per test and Vero B4 cells were used. The fourfold or higher difference in NT antibody titers of a particular serum between flaviviruses and here USUV and WNV is generally accepted as specific for the respective flavivirus [41]”.
R1: 4) The SNT results are difficult to follow. I suggest to the author include a different paragraph for each virus and include more information about USUV and how it was discarded from cross-reaction with other viruses.
AR: USUV was not in the primary focus of the study. It was the only potential flavivirus which might have caused cross reactions due to the close genetic relatedness to West Nile virus. As all sera were negative no cross reactivity could be detected and therefore no further mentioning of this virus was necessary. We added the following passage to the results of USUV (line 201-204):
“All the samples tested for USUV-SNT returned negative results. Therefore, this was prove that USUV was not responsible for the positive results and there was no cross reactivity in the tested sera due to this virus”.
R1: 5) Improve Figure 1.
AR: We have revised the Fig. 1. The figure has 300 dpi resolution.
R1: 6) Tables 3, 4, and 5 change to bold type the significant p-value.
AR: We have revised the tables as indicated.
Reviewer 2 Report
The work is a valuable contribution to revealing the seroprevalence frequency of selected Flaviviridae. Current knowledge on this subject is important due to a significant change in the way of life, but also climate change, which affects the spread of various viral diseases potentially dangerous to humans. The use of SNTs for the assessment of seroprevalence ratio, and not only testing the presence of antibodies by ELISA, significantly strengthens the power of the obtained results.
However, the manuscript requires a few methodological aspects to be clarified. Detailed comments are presented below:
- Lines 106-107
Please describe in detail how the number of participants in the study was reached. What were the inclusion and exclusion criteria for the study and on what basis was it determined that 200 samples from each county would be sufficient to obtain reliable results?
- Lines 109-110
On what basis the age ranges used in the study were adopted. Please justify the necessity to designate a relatively small group of participants aged ≥56.
- The results presented in chapter 3.3 are difficult to understand for the reader. Lines 186-188 Was SNT for WNV and TBEV performed on all positive and equivocal samples?
- In figure 1 the percentages for the bars describing ELISA-positive WNV are missing.
Author Response
Reviewer 2 (R2)
R2: The work is a valuable contribution to revealing the seroprevalence frequency of selected Flaviviridae. Current knowledge on this subject is important due to a significant change in the way of life, but also climate change, which affects the spread of various viral diseases potentially dangerous to humans. The use of SNTs for the assessment of seroprevalence ratio, and not only testing the presence of antibodies by ELISA, significantly strengthens the power of the obtained results.
However, the manuscript requires a few methodological aspects to be clarified. Detailed comments are presented below:
Lines 106-107
Please describe in detail how the number of participants in the study was reached. What were the inclusion and exclusion criteria for the study and on what basis was it determined that 200 samples from each county would be sufficient to obtain reliable results?
Author response (AR): The number of participants was established according to the budget allocated to the study: the maximum number of samples that could be processed according to the budget was 1200 samples.
Seroprevalence studies using blood donor study groups have limitations due to the exclusion and inclusion criteria as well as the social structure of the donors [https://www.who.int/publications/i/item/9789241548519]. The eligibility and exclusion criteria for blood donors in Romania is determined actually by the National Ministry of Public Health (Rule nr. 282 7 from 5th October 2005 - actualized on 9th April 2015 - regarding the admissibility of donors of human blood and blood components). Some examples for inclusion criteria: age between 18-65, body weight/temperature: ≥50 kg/<37.5°C, etc. More information regarding the exclusion and inclusion criteria here: https://legislatie.just.ro/Public/DetaliiDocumentAfis/167170.
R2: Lines 109-110
On what basis the age ranges used in the study were adopted. Please justify the necessity to designate a relatively small group of participants aged ≥56.
AR: There are several age categorizations. Donors were divided into age groups, broadly defined: covering the young adulthood, middle age and older adulthood. We used the same age categories as in our previously described studies. Since the same sera samples were tested also for Borrelia spp. IgG/IgM in previous study and for WNV/TBEV/USUTU in the present study, the statistical analysis was performed also for the same age categories as previously. In our previous study the age groups were divided according to Nancy M. Petry, 2002 (https://academic.oup.com/gerontologist/article/42/1/92/641498).
According to the blood transfusion centre guidelines in Romania, the maximum age of blood donors is 65. The old-adults group (≥56) is relatively smaller compared to young-adults or middle-aged because fewer blood donors usually show up annually at transfusion centres in Romania for several reasons (e.g., decrease in health after a certain age, thus failure to meet the eligibility criteria for blood donation).
R2: The results presented in chapter 3.3 are difficult to understand for the reader. Lines 186-188 Was SNT for WNV and TBEV performed on all positive and equivocal samples?
AR: We have revised the sentence.
R2: In figure 1 the percentages for the bars describing ELISA-positive WNV are missing.
AR: We have revised the Fig. 1.
Please see attachment.

Round 2
Reviewer 1 Report
I suggest the authors to make figure 1 including the abscissa and coordinate axes, and removing the lines.
Author Response
We have modified the picture as recommended.